# Arbovirus in Solid Organ Transplants: A Narrative Review of the Literature

**DOI:** 10.3390/v16111778

**Published:** 2024-11-15

**Authors:** Kiran Gajurel, Reshika Dhakal, Stan Deresinski

**Affiliations:** 1Division of Infectious Diseases, Carolinas Medical Center, Atrium Health, Charlotte, NC 28204, USA; 2Labcorp, Indianapolis, IN 46214, USA; reshika.dhakal@labcorp.com; 3Division of Infectious Diseases and Geographic Medicine, Stanford University, Stanford, CA 94305, USA; polishmd@stanford.edu

**Keywords:** arbovirus, chikungunga, dengue, donor derived, Japanese encephalitis, Powassan, yellow fever, tick borne encephalitis virus, West Nile, zika

## Abstract

The incidence of arbovirus infections has increased in recent decades. Other than dengue, chikungunya, and West Nile viruses, the data on arbovirus in solid organ transplant (SOT) are limited to case reports, and infections in renal transplant recipients account for most of the reported cases. Dengue and West Nile infections seem to be more severe with higher mortality in SOT patients than in the general population. Acute kidney injury is more frequent in patients with dengue and chikungunya although persistent arthralgia with the latter is less frequent. There is no clear relationship between arboviral infection and acute cellular rejection. Pre-transplant screening of donors should be implemented during increased arboviral activity but, despite donor screening and negative donor nucleic acid amplification test (NAT), donor derived infection can occur. NAT may be transiently positive. IgM tests lack specificity, and neutralizing antibody assays are more specific but not readily available. Other tests, such as immunohistochemistry, antigen tests, PCR, metagenomic assays, and viral culture, can also be performed. There are a few vaccines available against some arboviruses, but live vaccines should be avoided. Treatment is largely supportive. More data on arboviral infection in SOT are needed to understand its epidemiology and clinical course.

## 1. Introduction

Arboviruses are a heterogenous group of RNA viruses that are maintained in nature in various vertebrate hosts via transmission by arthropods. These viruses generally belong to *Togaviridae*, *Flaviviridae*, *Bunyaviridae*, *Reoviridae*, and *Orthomyxoviridae* family of viruses. A small number of them are known to cause human infections (Table 1). In recent decades, arboviruses like dengue, chikungunya, zika, West Nile, and oropouche virus have re-emerged and spread intercontinentally and established new ecological niches causing widespread human infection in areas not known to be endemic for these infections as a result of human activities as well as vector expansion, possibly related to climate change [1,2,3,4]. In addition, organ transplantation is being performed with increasing frequency in low- and middle-income countries, some of which are sites of frequent arobiviral infection. It is estimated that 3.9 billion people live in areas where dengue, yellow fever (YF), zika, and chikungunya virus transmission has occurred [5]. Human infections can be sporadic or occur in outbreaks depending on the vector activity and human interaction with the vector. When introduced into new environments where appropriate vectors exist, these viruses can become established and be transmitted autochthonously. This raises the specter of further spread of these viruses into new territories afflicting the human population. There are limited data on arbovirus infections in solid organ transplant (SOT) recipients [6,7]. In this review we explore the epidemiology, clinical features, diagnosis, pre-transplant screening, prevention, and treatment of clinically significant arboviruses in SOT recipients (For methods see Appendix A).

## 2. Medically Significant Arboviruses in SOT

### 2.1. Dengue

Dengue infection is caused by one of the four serotypes of dengue virus (DENV), a flavivirus, which is endemic in tropical and subtropical countries. In recent years autochthonous transmission of dengue has been witnessed in previously unaffected areas including continental United States and Europe [1,8,9,10]. Worldwide there has been a 10-fold increase in dengue infection from 505,430 cases in 2000 to 5.2 million in 2019 [11]. Since most infections are asymptomatic, the actual annual number of dengue cases is estimated to be much higher, around 400 million [12]. Moreover, half of the world population is at risk of receiving a dengue infection. This puts SOT recipients living or traveling to endemic areas at risk of acquiring dengue infection [13].

#### 2.1.1. Epidemiology

DENV is usually transmitted by *Aedes aegypti* and *Aedes albopictus* species mosquitoes. Other modes of acquisition include blood transfusion, transmission via hematopoietic cell and organ donation, mucocutaneous exposure including needle stick injury, intrapartum/perinatal transmission, and possibly via breast feeding [14,15,16].

Most cases of dengue infection in SOT have been reported in kidney transplant (KT) recipients [17,18,19,20,21,22,23,24]. Dengue in SOT can occur year-round in endemic areas and any time after transplant [25,26]. In a retrospective study in Brazil, there were only two (0.1%) cases of Dengue among 1754 liver and KT recipients between 2001 and 2006 [18]. The incidence, however, may, of course, dramatically increase during outbreaks [27,28]. Although currently available dengue vaccination is not recommended in SOT, its implementation in routine immunization in high transmission settings is predicted to lower the incidence of dengue infection (especially DENV1 and DENV2 serotypes) in the general population and perhaps in SOT as well [29]. The true incidence of dengue in SOT is not known since almost all dengue literature in SOT are retrospective in nature that included only patients that sought medical care. It is not clear whether SOT recipients are more susceptible to dengue infection compared to the general population but dengue incidence in SOT parallels the trend in the community [30].

#### 2.1.2. Clinical Features

It is estimated that 75% of dengue infections worldwide are clinically inapparent or minimally symptomatic [12]. In the general population, the first episode of dengue among symptomatic patients includes fever, headache, retro-orbital pain, myalgia, arthralgia, nausea, emesis, and rash. Infection by one of the four serotypes does not protect against subsequent infection by the others. Subsequent infections can be severe and can be complicated by shock and hemorrhage in both the general and SOT population [27,31]. In a study of 102 KT recipients with dengue, severe symptoms including shock and hemorrhage were numerically more frequent in secondary vs. primary infection (16% vs. 7%) [27]. However, in the SOT literature, primary and secondary infections are often not differentiated and in some studies they are presumed to be primary or secondary without serological confirmation of prior infection [17,20,21,30,32,33,34,35].

In general, clinical features of dengue infection in SOT are similar to those of the general population. In a review of 168 KT recipients with dengue, fever (86% vs. 99%), headache (35% vs. 96 %), myalgia (47% vs. 92%), and arthralgia (20% vs. 76%) were found to less common, whereas pleural effusion (17% vs. 2%) and ascites (35% vs. 1%) were more common compared to a historical cohort of general population [26]. In another study of KT recipients, retro-orbital pain, conjunctival redness, thrombocytopenia on admission, and absence of arthralgia were more frequently encountered in patients with dengue [28]. Pleural effusion and ascites have been reported in several studies in SOT [17,27,32,33,35,36,37,38]. Unusual syndromes like colitis, myocarditis, pericardia effusion, cholecystitis, and hemophagocytic lymphohistiocytosis syndrome have also been described [35,39,40]. Bleeding at allograft sites complicated by cardiac tamponade and graft nephrectomy have been reported in severe cases [22,32,36,41,42]. Arthralgia, however, seems to be distinctly uncommon in SOT, perhaps due to immunosuppressive effect of antirejection medications [25,30,35,38,43].

Cytomegalovirus coinfection has been rarely reported [27,30,34,44]. Other coinfections can also occur [27,28,37,45]. In the study by Nazim et al., bacterial coinfections (including bacteremia, pneumonia, urinary tract infection, lung abscess) occurred in 17%, malaria in 5%, and fungal in 3% of KT recipients [27]. Patients with bacteremia were more likely to have longer hospitalization, severe disease, and death. In fact, in all seven patients who died, death was attributed to bacterial infection.

Thrombocytopenia, leukopenia, and transaminitis are commonly encountered in SOT with dengue infection [25,33,35,37,38,46]. The duration of thrombocytopenia seems to be more prolonged in SOT patients when compared to the general population [27,37,38,47]. In one small study, thrombocytopenia recovery seemed to correlate with serological response [46]. Antimitotic drugs liked mycophenolate used for the prevention of rejection can cause cytopenia but in the study by Nasim et al. antimitotic agents had no effect on the duration and severity of thrombocytopenia [27].

Transient acute kidney injury (AKI) is common, occurring in more than 50% cases in most studies [25,26,28,31,34,35,38,43]. AKI is more frequently encountered in KT recipients compared to non-transplant patients with baseline normal renal function [47]. However, most of them seem to recover renal function and renal allograft failure is uncommon [30,35,47]. In the largest study of dengue in KT recipients, renal allograft failure occurred in only 6.5% patients [26]. Renal function is less likely to recover to baseline in severe dengue [27]. Acute cellular rejection is uncommon [22,31,34,41]. One study from India, however, reported 2 (6%) cases of acute cellular rejection among 31 dengue patients [28].

Some studies have reported an overall favorable outcome with no mortality, [25,28,31,35,44,46] but severe dengue, sometimes fatal, is not uncommon [27,30,33,34,38,43]. In the study by Ribeiro et al., 2 (11%) out of 19 KT recipients, and in another study from South America, 7 (35%) of 20 SOT, developed severe dengue but without reported deaths [31,35]. Although overall mortality is generally less than 10%, it seems to be higher than that of general population with dengue infection [27,30,43]. In a systematic review of 168 KT recipients, severe disease and mortality rates (16% vs. 4% and 9% vs. 0.06%, respectively) appeared to be higher compared to a historical control of the general population [26]. Disease severity and mortality are not associated with gender or time since transplant, although mortality appears to be higher in the early post operative period [26]. The type of immunosuppression does not seem to affect mortality except perhaps cyclosporine containing regimens that seem to be protective against severe dengue in secondary cases [27,43]. New bleeding complications and ascites seem to be associated with the disease severity, graft loss, and mortality [26].

#### 2.1.3. Donor Derived Infection (DDI)

Possible DDIs have been reported in kidney, liver, and heart transplant recipients [19,21,22,23,24,28,32,36,45,46,48]. In some of these reports the evidence is largely circumstantial [21,22,24,28,32,36,45]. These studies also do not exclude the possibility of dengue transmission via blood product transfusion.

The strongest evidence of DDI was presented by Mathew et al. in a liver transplant recipient where there was near complete nucleotide and amino acid homology in the envelope region of donor and recipient DENV type1 [49]. However, since the same strain of the virus might be circulating in the community, it is difficult to establish DDI definitively even when there is genetic identity between the donor and recipient DENV. Moreover, the recipient’s pre-transplant blood was not available so absence of subclinical infection prior to the transplant could not demonstrated.

Donor transmission from aviremic donors has been reported in KT recipients, presumably via contaminated urine in which DENV may be recoverable for longer than in the blood [46,48]. In one study, two KT recipients from the same donor who had negative blood DENV PCR and NS1 antigen, developed DENV2 infection soon after transplant [48]. Both recipients had negative pre-transplant blood PCR and or serology. The donor’s urine, on retrospective analysis, was found to be positive for DENV2 by PCR but it could not be analyzed by further sequencing. However, genetic sequencing of DENV isolated from the donor’s husband, who had dengue infection around the same time as the donor, matched that of the recipients. The recipient of a liver from the same donor did not develop dengue, suggesting urine was the probable route of DENV transmission.

Detection of DENV by PCR in podocytes in a KT recipient suggests that the virus can persist and possibly replicate in the kidney and can serve as a transmission source [50]. In the general population, DENV PCR in the urine may remain positive longer than in the serum, consistent with transmission via the urine and urinary tract being a potential route of transmission to KT recipients, but detection of viral RNA does not necessarily indicate the presence of replicable virions. The incidence of detection of DENV by PCR in the serum is >50% on day 0–7 of infection but then decreases significantly, whereas in the urine the rate of positivity is >50% on days 6–16 [51]. Moreover, DENV has been detected by PCR and immunohistochemistry but rarely by culture in various organs including kidney, liver, and lung from fatal cases of dengue [17,52].

DDI usually occurs within the first 10 days after transplantation. The outcome of early infection is similar to that after later onset. The symptoms range from none but with laboratory abnormalities (some diagnosed during investigation of dengue in other recipients from the same donor) to severe and even fatal [22,32,36,45,46,48]. In contrast, in a review of DDI in liver transplantation, five out of six transplant recipients had severe dengue and two died [24].

#### 2.1.4. Diagnosis

The diagnosis can be confirmed by blood PCR and or NS1 antigen detection. Blood PCR and NS1 are generally positive in the first week after illness onset but sensitivity decreases considerably thereafter in the normal host [51]. However, in the immunocompromised host, blood and more specifically urine PCR can remain positive for weeks [45,46,48,50]. In one study, blood and urinary PCR remained positive up to 65 and 365 days, respectively [46]. Positive PCR does not necessarily equate with the presence of infectious virions and urinary viral isolation by culture is generally positive for only a few days [36,48]. However, in one report of a lymphocytopenic KT recipient, urinary DENV was isolated for as long as 9 months after infection suggesting persistent viral replication [50]. Clearance of viruria or urinary PCR seems to coincide with CD8T cell and or general lymphocyte recovery [46,50].

Detection of the IgM antibody against DENV is less specific due to cross reactivity with other viruses, but it remains positive longer in the normal host. Serum IgM is usually first detected a week after onset of illness. In SOT, serological response can be suboptimal or delayed [27,46,49]. In most studies of dengue in SOT, serum IgM and NS1 have been employed more often than PCR [25,34]. Ideally combination of serum DENV PCR, NS1, and serum IgM will maximize sensitivity. In one study serum PCR, NST1, and IgM were positive in 87%, 72%, and 84% of cases [34]. A positive IgM alone should be interpreted in a clinical context. When possible, a positive IgM should be confirmed by a neutralizing antibody using quantitative assays like plaque reduction neutralizing tests (PRNT). While PRNTs are less likely to be false positive compared to IgM tests, their specificity decreases in areas endemic for multiple flaviviruses. Infection with one flavivirus can induce multiple neutralizing antibodies against other flaviviruses too (especially in seconday infection), and this can hamper the identification of the infecting virus [53]. Ideally serology should be performed on both acute and convalescent samples to confirm recent infection.

#### 2.1.5. Pre-Transplant Screening

Pre-transplant screening for dengue in asymptomatic donors and recipients is controversial. Some centers in endemic areas have implemented routine donor and or recipient screening by blood PCR (and urine PCR among KT donors) [30,46,48]. In Singapore, a dengue endemic country, universal blood dengue PCR screening in deceased donors was implemented in 2016, and in 2021 urine PCR was added to the universal screening protocol [54]. Among 207 deceased donors screened between 2016 and 2022 after universal blood PCR was implemented, only 1 (0.5%) donor tested positive. Although no cases of DDI were recorded after universal donor screening was initiated, the positivity rate among screened deceased donors was exceedingly low. Some authors recommend donor screening only during outbreaks [31,55]. However, positive dengue tests in deceased donors, unlike SOT recipients, may not corelate with community outbreaks [30,54]. South American guidelines recommend screening donors for exposure risk and suggest blood PCR screening in areas with ongoing viral activity [56], and South Asian guidelines recommend donor and recipient screening with NS1 or IgM during increased disease activity in the community [57].

Routine screening in resource limited endemic areas can be difficult to implement, and blood/urine PCR can add to the cost of already overwhelmed resources. Moreover, a negative PCR and NS1 antigen does not rule out recent infection since the sensitivity of both tests decrease after one week of illness, and DENV can be present in various organs with potential for reactivation after the transplant [50]. Moreover, the NS1 antigen has suboptimal sensitivity in secondary dengue infections [58]. Serum IgM can be used in resource limited settings, but it can be a false negative early on and can cross react with other viral infections. It might be prudent for individual transplant centers to establish their own criteria for donor/recipient screening based on disease prevalence in the community and available resources. However, if a donor is known to have dengue within last 30 days, transplantation should be avoided [56,57].

#### 2.1.6. Prevention

Avoidance of mosquito bites using protective clothing, insect repellants, and mosquito nets (day time sleeping) is likely the most cost-effective preventive strategy. Integrated pest management that includes using larvicides and adulticides and reducing mosquito habitats is needed to control mosquitoes [59]. Although not yet recommended by the World Health Organization (WHO), deployment of laboratory infected *A. aegypti* mosquitoes by *Wolbachia*, an intracellular bacterium, seems to be promising in endemic areas to control dengue infection [60]. These infected mosquitoes have a reduced ability to reproduce and are less likely to carry and transmit arboviruses including DENV.

Two live attenuated tetravalent dengue vaccines have been approved by the WHO in dengue endemic areas. CYD-TDV (Dengvaxia) is no longer manufactured due to lack of demand and difficulty implementation in immunization protocols in endemic areas (since it required screening for evidence of prior dengue infection). It had previously been recommended for people aged 9–45 years by the WHO (6–16 years in dengue endemic United States territories) who had laboratory confirmation of prior dengue infection. TAK-003 (Qdenga) is the only available vaccine approved by the WHO (but not approved in the United States) for patients 6–16 years of age (and for all persons ≥ 4 years of age in Europe) during high dengue transmission settings. Unlike CYD-TDV, TAK-003 can be used even in dengue seronegative people and, hence, does not require screening for prior dengue infection, but its efficacy against DENV3 and DENV4 in seronegative individuals is unproven, and there is some concern of severe infection among those without prior infection if infected with a virus of these serogroups [29]. TAK-003 is a live attenuated vaccine and not recommended in SOT. It can, however, be considered in SOT candidates prior to transplant in endemic areas with high rates of dengue transmission, and certain at-risk travelers to endemic areas, especially those who are likely to have had prior dengue and at risk of having a severe dengue infection during travel [29,61].

#### 2.1.7. Treatment

Treatment is supportive. Although the type of immunosuppressants do not seem to affect the overall outcome in most studies, immunosuppressive medications have been decreased or modified, partly due to cytopenia [25,30,34,35,38]. There are a few reports of successful liver transplantation in immunocompetent patients with dengue related acute liver failure [62,63]. Dengue is, however, a systemic infection, and there is a concern that the infection might worsen after immunosuppression, and liver transplantation is not routinely recommended for dengue-related acute liver failure [64,65].

### 2.2. Japanese Encephalitis (JE)

JE, caused by JE virus (JEV), is endemic in Southeast Asia and the Western Pacific, especially in rural and agricultural areas. It is generally transmitted by Culex species mosquitoes. Most infected people are asymptomatic or have mild symptoms. Only less than 1% develop encephalitis with significant mortality and morbidity.

#### 2.2.1. Epidemiology, Clinical Features, and Diagnosis

JE has been rarely described in SOT and, thus, its clinical course in this population is not well known [66,67]. In one case, JEV was transmitted to a lung transplant recipient via blood product transfusion with a fatal outcome, while another immunocompromised patient who also received blood products from the same donor had only asymptomatic seroconversion [66]. Serum and cerebrospinal fluid (CSF) JEV IgM testing are the primary modes of diagnosis, but SOT patients may not be able to mount a serological response and, in addition, the commercially available JEV IgM test is non-specific and can cross react with other viruses. Confirmatory plaque reduction neutralization tests (PRNT) may be available in reference centers. Serum JEV PCR in the general population with suspected infection has poor sensitivity due to low grade, transient RNAemia [68]. However, it might be a useful adjunctive diagnostic tool in SOT as was seen in the fatal case of JE in a liver transplant recipient where JEV RNA was detected in serum, CSF, and bronchoalveolar lavage [66].

#### 2.2.2. Pre-Transplant Screening

Routine donor and recipient screening is not recommended in endemic areas due to the rarity of this infection in SOT as well as the possibility of having false positive screening test results. However, it can be considered during outbreaks [57]. Suboptimal sensitivity of PCR and cross reactivity of serology makes screening challenging.

#### 2.2.3. Prevention

Mosquito control: Integrated mosquito management programs (e.g., mosquito surveillance, reduction in mosquito breeding sites, use of larvicides and adulticides, community education, etc.) and avoidance of mosquito bites using protective clothing, insect repellants, and impregnated mosquito nets at night time are the most cost-effective preventive strategies.

In the United States, the inactivated Vero cell culture-derived JE vaccine is available for travelers (including SOT recipients) to endemic areas who are at increased risk of becoming exposed to JEV. Although its efficacy in SOT is not clear, the seroprotection rate in adults after the standard two-dose vaccination is generally greater or equal to >95%. Other types of JE vaccines including live attenuated, live recombinant, and mouse brain-derived vaccine are available in other parts of the world and are recommended to the at-risk general population. However, live vaccines should be avoided in SOT recipients.

#### 2.2.4. Treatment

Treatment is supportive.

### 2.3. Chikungunya

Chikungunya is a viral infection caused by Chikungunya virus (CHKV), and like DENV it is transmitted by *A*. *aegypti* and *A. albopictus* mosquitoes. Other modes of acquisition include transmission via blood products and intrapartum transmission. In the last two decades, CHKV has spread to newer areas of the world and has the potential to establish autochthonous transmission in areas where *A. aegypti* and *A. albopictus* are already present [69].

#### 2.3.1. Epidemiology

Its predilection and incidence in SOT are not known, and the latter is affected by its background prevalence and the occurrence of outbreaks. Nonetheless, it seems relatively rare in SOT, probably due to underreporting and since most cases of chikungunya have occurred in places where organ transplantation is relatively infrequent. Chikungunya in SOT was first mentioned in 2007 in three KT recipients among 610 patients who had atypical clinical features of chikungunya during an outbreak in the reunion island [70]. Since then, there have been scattered reports of chikungunya in SOT, predominantly in kidney and liver transplant recipients [71,72,73,74,75,76,77,78,79].

#### 2.3.2. Clinical Features

Unlike dengue, most infected patients in the general population are symptomatic and present with a sudden onset of fever, headache, myalgia, cutaneous rash and arthralgia, and/or arthritis after an incubation period of 2–12 days [69]. Chikungunya can occur any time after a transplant. In a series of 32 KT recipients from Brazil, the mean time from transplant was 27.5 months (2–307 months) [77]. Most clinical features were like that of the general population with fever reported in 88%, rash in 47%, conjunctival hyperemia in 28%, and headache in 75%. All patients had arthralgia, and almost half of them had arthritis. Arthralgia is usually symmetrical and involves multiple joints—predominately small and medium joints [73,77]. Compared to the general population, musculoskeletal pain, including arthralgia and arthritis, seems to be less common in SOT [73,77,79]. Atypical and severe features like pneumonitis, myocarditis, and encephalitis are uncommon in SOT [70,71,78]. Transient AKI is not uncommon, occurring in 21% KTRs in one study [73,77,79]. Leukopenia including lymphopenia, thrombocytopenia, and transaminitis can occur [73,75,77,79]. CHKV viral load in the blood can be more than 1 million copies/mL and many SOT patients can have comorbidities that can be associated with severe disease [75]. However, the clinical course in SOT is generally benign except for persistent arthralgia [73,74,75,77,79]. Hospitalization may be required for AKI and pain management for arthralgia [73,77]. Chikungunya does not seem to be associated with rejection [73,75,79].

#### 2.3.3. DDI

DDI has not been documented in SOT. However, infectious CHKV has been found in eye tissues of infected persons, and the spleen, liver, and muscle in non-human primates, suggesting a possibility of DDI from infected donors [80,81].

#### 2.3.4. Diagnosis

In the general population, serum/plasma PCR is usually positive in the first week of illness, and serum/plasma IgM and neutralizing antibodies are detected in the second week. However, SOT recipients may not be able to mount a serological response, thus, causing difficulties in diagnosis. Viral culture can also be performed in the first days of illness, but it may not be feasible in most settings.

#### 2.3.5. Pre-Transplant Screening

Although some centers in endemic areas may perform universal donor blood chikungunya PCR screening, routine pre-transplant screening of asymptomatic donors and recipients is not recommended (see Section 2.2) [54,56,57]. However, during increased disease activity in the community, plasma/serum PCR screening of donors can be considered (see dengue) [56]. If a donor is known to have chikungunya within the last 30 days, transplantation should be avoided [56,57]. Successful kidney transplantation from a donor after recovery from chikungunya and who subsequently had negative serum PCR has been reported [82].

#### 2.3.6. Prevention

Mosquito control: See dengue prevention (Section 2.1.6).

In November 2023, the first Chikungunya vaccine (Ixchiq) was approved in the United States for at-risk patients, but it is a live attenuated vaccine and not recommended for SOT recipients.

#### 2.3.7. Treatment

Treatment is supportive. Joint pain, which may be persistent after resolution of active infection, can be severe and may require steroids (or escalation of steroids if they are already on) for relief [73,74]. Methotrexate with or with leflunomide in combination with steroids seems to be effective in severe cases [83]. There are no specific data on the management of immunosuppressants during chikungunya infection.

### 2.4. Yellow Fever (YF)

Yellow fever virus (YFV), the etiologic agent of YF, is endemic in regions of Africa and Central and South America. In its natural habitat (sylvatic cycle), the virus circulates among non-human primates via the Aedes species in Africa and via Haemagogus and Sabethes species in South America. Humans become infected when they encounter these vectors in their natural habitat, and the virus can circulate among humans via the urban vector, *A. aegypti* (urban cycle). The vaccine virus has been documented to be transmitted via blood transfusion and breast milk [16,84]. Case fatality rates among those with severe disease can be 30–60%.

#### 2.4.1. Epidemiology and Clinical Features

There are only a handful published cases of YF in SOT, making it difficult to generalize clinical findings [85,86,87]. Cases present with non-specific fever, malaise, and gastrointestinal symptoms [85,86,87]. Neurologic symptoms like unsteady gait, tremors, diplopia, nystagmus, myoclonus, and hemiparesis may develop [87]. Transaminitis and hyperbilirubinemia can be prominent. AKI and thrombocytopenia may develop. In severe cases, progressive encephalopathy and liver failure can occur.

#### 2.4.2. DDI

Transmission of vaccine strain YFV has been reported in four SOT recipients from a common deceased organ donor [87]. The organ donor had received a blood product donated six days after receiving the YF vaccine by a third party. Three days after receiving blood transfusion, organs were procured for transplantation. The four organ recipients (two kidneys, a heart, and a liver) developed symptoms within 6 weeks of transplantation. All of them developed significant neurologic symptoms and two of them eventually died. YFV was detected in all recipients by either PCR, metagenomics, or serology (CSF or serum). YVF detected in the CSF in a KT recipient and brain tissue in heart transplant recipient was identical or similar to the vaccine strain on sequencing. The organ donor, however, tested negative for YFV, likely due to dilutional effect of fluids received during treatment or the presence of only low-level viremia.

#### 2.4.3. Diagnosis

In infection in the general population, serum viral RNA may be detected in the first 4 days of illness. Viral culture (blood) can also be positive early in illness but may not be feasible in most clinical settings. YFV can, however, be detected for several days by culture or RT-PCR in other body fluids like urine and semen during convalescence even when serum PCR is negative [88]. Serum YFV-specific IgM testing can be used later in illness, but its specificity is limited by cross reactivity with other flaviviruses. Moreover, YF vaccine recipients can have persistently positive serum IgM for several years. Whenever possible, a positive serum YF IgM should be confirmed by a more specific PRNT. In SOT recipients with severe disease, YF-specific IgM and metagenomics have been used in CSF for diagnosis [87]. Real time-PCR and immunohistochemistry in tissue specimens can also be helpful in establishing disseminated disease [86,87,89].

#### 2.4.4. Pre-Transplant Screening

Routine pre-transplant screening is not performed (see Section 2.2). The donors should be screened for recent exposure or travel to endemic areas. During an epidemic, serum YFV PCR screening can be considered [56]. Organs from persons with YF disease and recent YF vaccination should be avoided for 30 days, and blood donation for YF vaccine recipients should be avoided for at least 2 weeks [56,87].

#### 2.4.5. Prevention

Mosquito control against urban YF: See Dengue prevention (Section 2.1.6).

Live attenuated YF vaccine is available for prevention of YF infection. Although (inadvertent) YF vaccination in SOT recipients several months or years after transplant seems to be safe and immunogenic, it is generally contraindicated in this population [90,91,92,93,94,95]. Vaccinated patients who later undergo SOT can maintain serological immunity even years after transplant despite immunosuppression [96,97]. Hence, whenever possible, at-risk patients should be vaccinated prior to transplantation. Due to an increased risk of YF-associated viscerotropic and neurologic disease, the benefits and risks of vaccination should be weighed carefully when vaccinating patients ≥ 60 years.

#### 2.4.6. Treatment

Treatment is supportive. There are anecdotal cases of successful liver transplantation in non-transplant patients with liver failure from YF [89,98,99]. However, YF is a systemic disease affecting various organs, and YFV can infect the engrafted liver [89]. In one report from Brazil, only 6 (20%) of 30 non-transplant patients who underwent liver transplantation for liver failure survived [99]. Hence, liver transplantation cannot be routinely recommended for YF-related liver failure.

### 2.5. Zika Virus

Zika is a systemic infection caused by Zika virus (ZKV). It is primarily transmitted by *A. aegypti* and *A. albopictus* in endemic areas. Other modes of transmission include perinatal transmission, and transmission via sexual contact and blood products. Transmission via animal bites, needle stick injury and saliva, urine, and breast milk (despite the presence of replicative ZKV RNA in various body fluids) is not well established [16,100].

#### 2.5.1. Epidemiology

ZKV can establish autochthonous transmission in areas with no prior history of zika infection [3,101]. When ZKV spread to South America in 2014, there was widespread apprehension that it would be an especial problem among potential organ donors and recipients [102,103]. Surprisingly there have only been five case reports of zika in SOT to date [104,105,106]. The first ever case series in SOT described two liver and two KT recipients with zika infection 43–590 days after transplant [105]. The mode of acquisition is not clear. In one report, a liver recipient received ZKV infection via infected platelet transfusion on the day of transplant from a blood donor who developed symptoms three days after blood donation [104]. The recipient remained asymptomatic but replicative viral RNA was detected in the serum, which, on genomic sequencing, matched closely with viral RNA in archived blood of the donor. In another report, a heart transplant recipient developed fatal ZKV infection eight months after the transplant [106].

#### 2.5.2. Clinical Features

In the general population, most of the infections are asymptomatic. Symptomatic patients may develop self-resolving fever, rash, headache, arthralgia, myalgia, and conjunctivitis. Neurologic symptoms like Guillain–Barré syndrome and congenital zika syndrome (especially microcephaly in a newborn) have also been reported.

In the published literature on SOT, the symptoms have ranged from asymptomatic to fatal [104,106]. Fever and myalgia seem to be common, but rash and conjunctivitis were not described in the published cases. In the fatal case of zika infection in a heart transplant recipient, the patient presented with fever, headache, malaise, hemiplegia, and seizure [106]. The patient was eventually found to have meningoencephalitis. MRI of the brain showed hypo- and hyperintense lesions in the cingulate and superior frontal gyrus. CSF analysis showed lymphocytic pleocytosis with elevated protein and positive ZKV PCR in the CSF. Immunosuppression was reduced, but the patient had progressive neurologic decline and developed refractory shock secondary to allograft rejection. On autopsy ZKV was detected in various organs including the brain, heart, liver, and lung by PCR, immunofluorescence, and or electron microscopy. In the case series of four SOT with zika, all were hospitalized and had bacterial co-infection. Two of them had fever and three patients had myalgia. All had thrombocytopenia and the liver recipients developed transaminitis while the KT recipients developed AKI. One liver transplant recipient required re-transplantation three months later due to hepatic artery thrombosis and biliary stenosis, which may or may not be related to ZKV. None had neurologic symptoms and all survived.

#### 2.5.3. DDI

Although ZKV transmission from a blood donor to a liver recipient has been reported, no ZKV transmission from an organ donor has been described [104]. In one study, two kidneys from a donor with positive ZKV serum IgG (but negative serum IgM and negative serum and urine PCR) were transplanted [107]. None of the KT recipients developed zika infection. However, ZKV has been found in various body fluids in infected people. ZKV has also been shown to infect proximal tubular epithelial cells, glomerular podocytes, and endothelial and mesangial cells [108,109,110]. ZKV RNA is found to persist longer in the urine than serum PCR and replication competent virus has been isolated from urine [111]. Thus, the kidneys can serve as reservoir of ZKV and can potentially transmit infection via a renal allograft.

ZKV can also replicate in human cornea, albeit less efficiently, and ZKV RNA has been found in aqueous and vitreous humor and a conjunctival swab, but there has been no documentation of DDI via a corneal graft [112,113,114]. In one study a deceased donor with negative serum ZKV RNA was found to have ZKV RNA in the vitreous humor after cornea transplantation in two patients [115]. None of the recipients developed zika infection.

#### 2.5.4. Diagnosis

Serum and urine ZKV PCR can be used for diagnosis in early infection. In the general population, serum PCR is usually positive in the first week after symptom onset and urine PCR remains positive for 2 weeks although a longer duration of RNA in serum and urine has been reported [116]. Whole blood PCR seems to be more sensitive than serum [117]. Viral culture can be performed in early infection but may not be pragmatic. Serum IgM is generally positive after the first week of infection and can persist for several weeks. But IgM can cross-react with other viruses and can be false positive [118]. Whenever possible PRNT should be performed to confirm a positive serum IgM, but PRNT may not be able to confirm whether the current infection is from ZKV or a recently exposed flavivirus like dengue, especially when there is cocirculation of ≥2 similar arboviruses in the community [53]. Zika PCR can also be used in other body fluids or organs like the brain, CSF, and aqueous and vitreous humor [106,112,113,114,115]. Immunofluorescence and Immunohistochemistry can also be used to detect ZKV in tissues [106].

#### 2.5.5. Pre-Transplant Screening

Routine testing of donors and recipients for ZKV infection is not recommended (see Section 2.2), although some centers in endemic areas may perform universal donor blood PCR screening [54]. During epidemics, routine serum or blood PCR monitoring can be considered [56]. In non-endemic areas, donors (and recipients) who have had possible exposure to ZKV due to travel or sexual contact should be tested for ZKV infection [56,116]. The ideal way to screen for ZKV is not clear since serum/blood PCR is only transiently positive, while a positive serum IgM test result can linger on for months. Organs from asymptomatic donors with negative plasma/blood and or urine PCR but positive serum IgM and IgG can be accepted after weighing the risk–benefit ratio [107,118]. Organs from donors with zika infection should not be accepted for 120 days [57].

#### 2.5.6. Prevention

Mosquito control: See dengue prevention (Section 2.1.6).

SOT recipients should avoid unprotected sex with a male partner potentially exposed to ZKV for 3 months after the return of the male partner from an endemic area or after symptom onset [119]. Similarly, male SOT recipients should avoid unprotected sex with a female partner potentially exposed to ZKV for 2 months after return of the female partner from an endemic area or after symptom onset [119]. Pregnancy should be avoided during this time.

#### 2.5.7. Treatment

Treatment is supportive.

### 2.6. Powassan Disease

Powassan virus, the etiologic agent of Powassan disease, is endemic in the United States (especially in Northeast and around the Great lakes area), Canada, and Russia. It is transmitted by the bite of infected Ixodes species (especially *I. sacpularis*) and less commonly via blood transfusion.

#### 2.6.1. Epidemiology

Descriptions of Powassan disease in SOT are limited to case reports. One report described Powassan virus transmission in a KT recipient in the immediate post-transplant period via blood transfusion from an asymptomatic donor with history of tick bites [120]. The other case report described Powassan disease, presumably contracted by a tick bite, in a KT recipient 14 years after transplant [121].

#### 2.6.2. Clinical Features

Infection in the general population is mostly asymptomatic. It can, however, cause encephalitis with 10% mortality among those with severe disease and with significant morbidity among survivors. It has a long incubation period of 1–4 weeks. In a SOT population there are very little data on the clinical course of Powassan disease. The initial signs and symptoms can be non-specific fever, headache, myalgia, and diarrhea with progression to neurologic symptoms including encephalitis [120,121]. CSF may show lymphocytic pleocytosis and elevated protein. Brain MRI may show T2 enhancement of the brainstem and the cerebellum. In the two cases described in the literature, both patients survived with residual neurologic deficit [120,121].

#### 2.6.3. Diagnosis

Blood and serum PCR can be positive in the early stage of the disease. PCR can also be performed in the CSF and formalin fixed tissues. A positive serum and CSF IgM test supports the diagnosis, but whenever possible a positive IgM test should be confirmed by a PRNT. Immunohistochemistry can also be performed on fixed tissue specimens.

#### 2.6.4. Pre-Transplant Screening

Routine pre-transplant screening is not recommended (see Section 2.2). Organs and blood products from donors with Powassan disease should be avoided for at least 4 months.

#### 2.6.5. Prevention

In endemic areas, preventive measures should be taken to avoid exposure to ticks. There is no vaccine available against Powassan disease.

#### 2.6.6. Treatment

Treatment is largely supportive. The role of steroids and intravenous immunoglobulin (IVIG) is not clear.

### 2.7. Tick Borne Encephalitis (TBE) Virus

Tick-borne encephalitis virus (TBEV) is endemic in parts of Europe and Eastern Asia and causes TBE. There are three different stains of TBEV—European, Siberian, and Far Eastern; the latter two have a worse outcome. It is primarily transmitted by Ixodes species and via ingestion of contaminated dairy products. Other modes of transmission include handing of infected material (possibly via aerosolization and possibly via percutaneous and mucosal exposure), slaughtering viremic animals, breast feeding, blood transfusion, and organ transplantation [122,123].

#### 2.7.1. Epidemiology

TBE is uncommon in SOT. In a Swiss study involving 4967 SOTs during 2008–2019, there were only 2 cases of TBE, with an incidence rate of 0.09/1000 person-year [124].

#### 2.7.2. Clinical Features

In the general population, most infected patients are asymptomatic. Among symptomatic patients, the illness can be biphasic with an initial febrile syndrome followed by an asymptomatic period and then progression to neurological symptoms including encephalitis.

Based on limited case reports, TBE can have a worse outcome in SOT. Fatal encephalitis has been reported in transplant recipients [125,126,127]. The symptoms start with a febrile illness and progress to neurologic symptoms. These patients may have a monophasic illness without an asymptomatic period in between febrile illness and neurologic symptoms [126]. CSF can be normal or demonstrate lymphocytic pleocytosis with elevated protein levels. Brain MRI may show T2 hyperintensity in the brain stem and the cerebellum

#### 2.7.3. DDI

A cluster of fatal DDI was reported in three SOT recipients who received organs from the same donor [126]. The incubation period of 17–49 days in these patients seems to be longer than median incubation period of 8 days in the general population. The donor lived in an endemic area. The genomic sequencing of donor and recipients’ viral RNA confirmed the same viral strain.

#### 2.7.4. Pre-Transplant Screening

Routine pre-transplant screening is not recommended (see Section 2.2).

#### 2.7.5. Diagnosis

Blood or serum PCR can be performed in the early stage of the disease. PCR also be conducted on urine, CSF, and brain tissue [125]. Next generation sequencing has also been conducted on the brain tissue and CSF [126]. A positive IgM in the sera and CSF is suggestive of infection but whenever possible should be confirmed by PRNT.

#### 2.7.6. Prevention

In endemic areas, preventive measures should be taken to avoid exposure to ticks and consumption of unpasteurized dairy products. An inactivated vaccine against TBEV is available in the Unites States and is recommended for at-risk travelers to endemic areas [123]. It is also recommended for at-risk laboratory workers. Its effectiveness after a 3-dose series against the European strain is >90%, although the effectiveness can be diminished in SOT recipients. It should be noted that there are other local vaccines available in endemic areas.

#### 2.7.7. Treatment

Treatment is supportive. There is no clear role of IVIG (including TBEV specific IVIG) [123].

### 2.8. West Nile Virus (WNV)

WNV is transmitted to humans through the bite of Culex mosquitoes and accounts for the majority of mosquito-borne illnesses in the continental United States [128]. Transmission also occurs via blood transfusion, organ transplantation, intrauterine exposure, breast feeding, and percutaneous injury [4,129]. Autochthonous transmission with new areas of endemicity can occur in previously unaffected geographic areas [4].

#### 2.8.1. Epidemiology

WNV infection peaks during increased mosquito activity. During 2009–2018, approximately 90% of reported WNV illnesses in the United States occurred between July and September [130]. SOT recipients are at a higher risk of severe WNV disease [130,131]. Transfusion-transmitted infection can occur after receiving infected blood products directly by the SOT recipients or indirectly from allografts from organ donors who received blood products prior to organ procurement [132,133,134]. KT recipients account for more than half of WNV infection, likely reflecting a larger volume of kidney transplantation [135].

#### 2.8.2. Clinical Features

In the general population, 80% WNV infections are estimated to be asymptomatic, and most symptomatic patients have mild febrile illness with body aches, headache, gastrointestinal symptoms, and rash. Only <1% develop neuroinvasive disease (encephalitis, meningitis, and acute flaccid paralysis), although its incidence increases with age (0.02 vs. 1.22 cases per 100,000 population among aged < 10 years vs. ≥70 years, respectively) [130]. Case fatality rate also increases with age and is higher in those with encephalitis (14%) and acute flaccid paralysis (13%) vs. those with meningitis alone (2%).

Although SOT recipients can have asymptomatic or mild illness, they are at a higher risk of having neuroinvasive disease. During a 2002 WNV outbreak in Canada, the incidence of neuroinvasive WNV was found to be 40 times higher in SOT compared to the general population [131]. The increased risk was probably overestimated due to the small number of infected SOT patients. Nonetheless, in a review of 53 cases of WNV in SOT between 2002 and 2019, 48 (91%) patients had neuroinvasive disease (mostly encephalitis or meningoencephalitis) [135]. Another review of 69 cases of WNV in SOT also found neuroinvasive disease in 61 (88%) patients [133]. Although it should be noted that most milder and asymptomatic cases likely do not get reported.

In a review of 52 published cases of WNV in SOT, the median time to infection from transplant was 14 months, and the mean age of patients was 50 years [135]. The mean time to diagnosis from symptoms onset was 5.2 days. Presenting symptoms in SOT can be non-specific even in patients who eventually develop neurological illness [136,137,138,139]. In a study of 24 neuroinvasive infections in SOT, 88% and 71% of patients presented with gastrointestinal symptoms and fever, respectively [138]. Only 4 (17%) had cognitive impairment and 2 (8%) had acute flaccid paralysis at presentation. The median time to clinical worsening after admission was 4 days (range 1–11 days). Abnormal movement including myoclonus and parkinsonian features can occur in some patients [135,139]. In a retrospective study of neuroinvasive disease, immunocompromised patients were less likely to have headache and myalgia and more likely to have myoclonus and encephalopathy compared to non-immunosuppressed patients [140].

There is no clear association of WNV with acute cellular rejection, however, rejection may result from a reduction in immunosuppressants following infection (118, 122, 125). There can be slight decrease in renal function in KT recipients with WNV infection, but graft loss attributable to WNV is uncommon [141]. In one study of SOT, allograft loss occurred in 2 (4%) of 52 patients [135]. In the same study, the overall mortality was 37%. In another review of SOT with WNV, 18 (31%) out of 59 patients with known outcome died [133]. All patients who died had neuroinvasive disease where 18 (33%) out of 55 patients with known outcome died. This mortality rate is higher than that of general population where the overall mortality rate associated with neuroinvasive disease is 9% [130]. Other studies have also shown higher mortality in neuroinvasive disease although mortality seems variable in smaller studies [134,138,139,141,142]. In general, higher mortality is seen in immunocompromised patients [143]. A recent study of neuroinvasive disease also showed longer duration of hospitalization and higher rates of ICU admission, mechanical ventilation, and 90-day all-cause mortality in immunocompromised patients compared to non- immunocompromised patients [140]. As in the general population, mortality is higher among patients with acute flaccid paralysis and encephalitis than meningitis alone [138]. Neurologic deficit, including permanent damage, can be significant among survivors [135,138,139,141].

In neuroinvasive disease, brain MRI can show punctate subacute infarcts and T2 flair hyperintensity signals in the brain stem, thalamus, cerebellum, and mesial temporal lobes [137,138,139,140]. CSF can be abnormal with mild pleocytosis and elevated protein [137]. CSF white blood cell count tends to be higher in immunocompromised patients [140]. The CSF pleocytosis is neutrophilic predominant early on before transitioning to lymphocytic predominant [138]. The CSF glucose in generally within normal range.

#### 2.8.3. Diagnosis

WNV nucleic acid amplification test (NAT) or PCR can be performed in blood, CSF, and tissue. Next generation sequencing can also be performed in CSF. In general, WNV viral load is higher, and the viral RNA persists longer in whole blood compared to plasma [144]. The presence of nucleic acid in body fluids can be transient and WNV IgM (preferably with confirmatory PRNT) in blood and CSF is relied on for diagnosis at a later stage. In a study of 24 SOT with neuroinvasive disease, serum and CSF WNV IgM was positive in 63% and 48%, respectively, in whom it was tested [138]. In total, 8 (89%) out of 9 patients had positive serum NAT, while 8 (38%) out of 21 patients had a positive serum PCR. CSF PCR was positive in 44% of tested patients. In another review of SOT with WNV infection, serum RNA was positive in 10 (83%) out of 12 patients and serum IgM was positive in 15 (83%) out of 18 patients [137]. In patients with neuroinvasive disease 6 (75%) out of 8 patients and 10 (71%) out of 14 patients tested positive for CSF WNV RNA and IgM. In one study of kidney and or pancreas recipients, 17 out of 19 patients mounted a serologic response within 2–4 weeks [136]. In patients with neuroinvasive disease, positive CSF PCR tends to be more frequent in immunocompromised patients [140]. It is important to note that serological and RNA positivity depends on the timing of the test in relation to symptom onset, and negative tests do not rule out diagnosis. Other diagnostic tests include metagenomics in CSF, immunohistochemistry, and PCR on tissue [137,145,146,147,148]. Viral culture is rarely performed.

#### 2.8.4. DDI

In a retrospective study of DDI in 207 KT recipients from 139 donors between 1948 and 2017, WNV accounted for 13 (6.3%) infections, of which 5 (38%) died [149]. In a review of donor-derived diseases by the Disease Transmission Advisory Committee between 2008 and 2017, WNV accounted for 5 (2%) out of 250 proven and probable pathogens transmitted from 244 donors [150]. Although the estimated donor derived proven and probable WNV occurred in only 0.17 per 100,000 SOT recipients between 2012 and 2017, sporadic clusters of donor-derived WNV continue to occur. Abbas et al., in their review, noted that DDI accounted for a quarter of the 53 published cases of WNV [135]. To date, there are no reported cases of transmission of WNV through living donors [134].

The first cases of DDI were reported in 2002 in four SOT recipients from a common organ donor who had received blood transfusion from an infected donor prior to organ procurement [148]. DDI occurs in the immediate post-transplant period with an incubation period of (median) 13 days, and initial symptoms can be non-specific [137]. Soto et al. recently reported 2 cases of DDI and reviewed 21 previously published cases with adequate information [134]. Two of the eight organ donors became infected via blood transfusion prior to organ procurement, and the rest were assumed to be infected via mosquito bites. Overall, 23 (85%) out of 27 SOT recipients from 10 infected donors acquired infection, and 16 (70%) developed neuroinvasive disease. Six patients died with an overall mortality of 26% (38% among patients with neuroinvasive disease), like that of non-DDI.

#### 2.8.5. Pre-Transplant Screening

All potential donors should be screened for signs and symptoms of and risk factors for WNV exposure (e.g., travel to an endemic area). In the United States, the Organ procurement and Transplantation Network ad hoc Disease Transmission Advisory Committee recommends pre-transplant screening of both living and deceased donors with plasma WNV NAT during periods of heightened WNV activity [151]. Year-round WNV screening can, however, increase false positive rates, and it is discouraged to avoid inappropriate discarding of organs. Either seasonal testing (July through October in the continental United States) or triggered testing during increased WNV activity in the community (where the donor has lived or traveled to) can be performed. In deceased donors, however, the result may not be available at the time of transplantation, and not all organ procurement centers routinely screen for WNV infection in deceased donors. In one survey only 39% of 46 centers screened for WNV infection in potential deceased donors [152]. In living donors, plasma WNV NAT should be performed within 7–14 days prior to transplantation [151]. If NAT is positive, then the transplantation should be deferred for 28 days after which plasma NAT should be repeated and WNV IgM checked. If NAT remains positive, organ donation should be deferred. If both NAT and IgM are negative (likely false positive initial NAT test) or NAT is negative and IgM is positive (recovery from WNV infection), organ donation can be considered.

In the review by Soto et al., 9 out of 10 donors who transmitted WNV did not undergo pre-transplant testing. The archived serum on retrospective testing was positive for viral RNA in only six donors. In four donors, the transmission occurred despite negative viral RNA in the sera. Hence, despite pre-transplant screening for viral RNA, disease transmission can occur. However, screening will at least eliminate the infected donors who test positive. Serum/plasma RNA can be falsely negative in potential deceased donors due to dilutional effect from resuscitation measures or low-level RNAemia below the threshold for detection. Moreover, WNV can be detected in various organs or tissues by PCR, immunohistochemistry, and viral culture despite clearance from the blood [129,145,146,153]. WNV RNA has also been detected in urine for years after recovery from infection suggesting that kidneys can be a reservoir for viral reactivation [154]. In one instance of DDI, WNV was isolated in culture from lymph node/spleen tissue despite negative RNAemia [137,146]. A positive serum IgM/PRNT is suggestive of viral clearance from the blood, but as reviewed by Soto et al., viral transmission occurred from three donors with negative serum RNA who had mounted an IgM response. Thus, potentially replicative virus harboring in organs cannot be detected in donors with negative plasma or serum NAT and positive serum IgM, and this creates a challenge in organ transplantation [155]. Hence, careful assessment of donors should be performed to minimize the risk of transmission. It should be noted that potential donors with WNV infection may not have encephalopathy, and confounding diagnoses in donors with encephalopathy might lead to classifying them as low risk for disease transmission [149,156].

#### 2.8.6. Prevention

Mosquito control: See Japanese encephalitis section on cullex mosquito prevention (Section 2.2.3).

No vaccines are currently available for WNV prevention.

#### 2.8.7. Treatment

Treatment is supportive. IVIG, hyperimmune globulin, steroids, and interferons have been used in SOT with WNV infection, but data supporting their efficacy are lacking [140,157,158]. Immunosuppressants are reduced in >90% cases in published literature but its effect on survival is not clear [135].

### 2.9. Other Arboviruses

There are scattered case reports of less common arbovirus infection in SOT.

A cluster of donor-derived Eastern equine encephalitis (EEQ) viral infection has been reported in three SOT recipients (heart, liver and lung) [159]. All three organ recipients developed encephalitis within a week of transplant. The clinical course in the heart transplant recipient was complicated by acute cellular rejection or myocarditis. All recipients were treated with IVIG +/− steroids. Only the lung transplant recipient survived with residual neurologic deficit. The donor likely acquired the infection via mosquito bite and pre-transplant stored donor serum tested positive for EEQ virus RNA.

A few cases of encephalitis caused by St Louis encephalitis (SLE) virus (acquired via blood transfusion in the post-transplant period or mosquito bite) with high morbidity and mortality has been reported in SOT [160,161].

Similarly, Jamestown Canyon virus has been reported to cause encephalitis in a heart and a liver transplant recipient 4 and 3 years after transplant, respectively [162,163]. The infection was presumably acquired via a mosquito bite. Both patients survived.

The Cache Valley virus has been reported to cause meningoencephalitis in a KT recipient in the immediate post-transplant period [164]. The infection was thought to be acquired via blood transfusion on the day of transplant from an infected blood donor.

A fatal case of Crimean–Congo hemorrhagic virus, confirmed by a positive viral RNA in the blood, has been reported in a liver transplant health care worker who acquired it while performing surgery on a presumably infected patient [165].

Heartland virus infection has been reported in a heart transplant patient 7 years after transplant, likely to have occurred via a tick bite [166]. The patient presented with fever, leukopenia with lymphopenia, thrombocytopenia, transaminitis, myositis, and encephalopathy, and they eventually recovered. The diagnosis was confirmed by a positive blood and serum PCR.

Rift valley fever has bene described in a KT recipient returning traveler from Mali [167]. The mode of acquisition remained undetermined, but the patient endorsed mosquito bites, ingestion of raw milk, and contact with animals—all known risk factors for rift valley fever virus transmission. The patient presented with non-specific febrile syndrome that progressed to encephalitis. The patient made a full recovery. The diagnosis was supported by positive serum and CSF serology and confirmed by positive urine and semen PCR that remained positive for several weeks.

Usutu virus is an uncommon arbovirus that has been reported to cause encephalopathy in a liver recipient [168]. The liver recipient had undiagnosed Usutu infection at the time of transplant. The diagnosis was made by positive plasma PCR and gene sequencing. The virus was also isolated in cell culture (which on sequencing was identical to the post-transplant sequencing) from stored pre-transplant plasma and on sequencing.

Although no infections in transplant recipients have been reported to date, the recent explosion of the Oropouche virus in regions of South America, particularly Brazil, raise concern going forward [169]. 

#### 2.9.1. Diagnosis

As with other arboviruses, diagnosis is made by serum, blood, CSF, and tissue PCR. Urine and semen RNA PCR can also be positive as was seen with rift valley fever viral infection in a KT recipient [167]. CSF metagenomics have also been used in diagnosis [164]. Viral culture is cumbersome but can be performed, especially, in early infection [164,168]. Gene sequencing can be used for confirmation [164,168]. Blood/serum and CSF IgM is supportive of the diagnosis and should be confirmed with neutralizing antibodies like PRNT. Tissue immunohistochemistry has also been used to aid diagnosis [159].

#### 2.9.2. Prevention

Mosquito control, avoidance of mosquito bites, and ingestion of unpasteurized dairy products are recommended. Ribavirin has been used for prophylaxis in those exposed to Crimean–Congo hemorrhagic fever virus [170]. An inactivated vaccine against the Crimean–Congo Hemorrhagic virus is available in Bulgaria, but its effectiveness is uncertain [170].

#### 2.9.3. Donor Screening

Routine donor and recipient screening is not recommended (see Section 2.2).

#### 2.9.4. Treatment

Treatment is supportive. Although IVIG, ribavirin, favipiravir, hyperimmune globulin, and interferons have been used with antecedental reports of success, there are no strong data to support their routine use [159,160,161,162,163,164,165,170]. It may be prudent to reduce immunosuppressants during active infection, but the role of this strategy in outcome is not clear [166].

## 3. Conclusions

The clinical course of arbovirus in SOT is generally more severe and carries a higher mortality compared to that of the general population. However, the data on arboviruses are limited, and asymptomatic and milder illnesses likely go undetected. Diagnostic tests, like PCR and NAT, are transiently positive in most cases, and IgM tests suffer from poor specificity. Confirmatory neutralizing antibodies like PRNTs are not widely available. Even PRNT may not be able to distinguish the causative viral agent from another recently exposed similar arbovirus. Ideally paired acute and convalescent specimens are required for serological testing to confirm acute infection by demonstrating seroconversion and a fourfold rise in titer, but this may not be pragmatic and does not provide diagnosis in real time. Moreover, most of these assays may not be available in resource-limited settings. SOT recipients also face a unique situation where they can acquire infection from donor allografts as well as blood donors in addition to de novo infection caused by exposure to the vector during community outbreaks of arboviruses. This poses a challenge to transplant centers in selecting and accepting appropriate donor organs. While implementation of screening strategies in organ donors to prevent transmission of virus to transplant recipients is necessary, overzealous screening and imperfect diagnostic tests can lead to false positive tests and discarding of potentially lifesaving organs.

## 4. Future Direction

More data on arbovirus infection in this population and better diagnostic tests and effective therapeutics are needed.

## Figures and Tables

**Table 1 viruses-16-01778-t001:** Medically significant arboviruses.

***Togaviridae* family:**
Eastern equine encephalitis virus
Western equine encephalitis virus
Venezuelan equine encephalitis virus
Chikungunya virus
Sindbis virus
Ross river virus
Barmah Forest virus
***Flaviviridae* family:**
Dengue virus
Japanese encephalitis virus
Murray valley fever virus
St Louis encephalitis virus
West Nile virus
Powassan virus
Tick Borne encephalitis virus
louping ill virus
Omsk hemorrhagic fever virus
Alkhurma Hemorrhagic Fever Virus
Mayaro virus
Kyasanur Forest Disease Virus
Zika virus
Yellow fever virus
Usutu virus
Rocio encephalitis virus
***Bunyaviridae* family (Bunyavirales)**
California encephalitis virus
-La Crosse encephalitis virus
- Jamestown Canyon
Rift Vallet fever virus
Crimean Congo hemorraghic fever virus
Severe fever with thrombocytopenia syndrome
Heart land virus
Oropouche Virus
Toscana Virus
Cache Valley fever
Ngari Virus
***Reoviridae* family**
Coltivirus
-Colorado tick fever virus
-Seadornavirus (Banna virus)
***Orthomyxoviridae* family**
Thogotovirus genus including thogoto, bourbon and dhori viruses

## Data Availability

Not applicable.

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
