# Peer review of "Arbovirus in Solid Organ Transplants: A Narrative Review of the Literature"

_viruses, 2024, doi:10.3390/v16111778_

Round 1
Reviewer 1 Report
Comments and Suggestions for Authors
Dear Authors,
The work entitled “Arbovirus in solid organ transplant” provides an extensive revision on important aspects of the dramatic events that arise when any arbovirus infection occur during organ transplantation. Nevertheless, I feel that the extension and the broad scope of the review will be overwhelming for the reader of Viruses (still for the specialized reader working on the topic transplantation). Accordingly, I suggest some changes to make it more clear, dynamic and understandable as well.
Suggestions:
0) Please shorten as much as you can the length, maintaining only the most important information, if possible.
1) In table 1 it will be interesting to incorporate information on which arbovirus cross-react with which (by ELISA, PCR, etc)? It will be interesting to add to this Table 1 in order to detect the infection in the patient and guess which virus is present based on geographical area or recent travels.
2) In line 83 please generalize to Aedes mosquitoes in endemic areas, instead of citing species.
3) In lines 93 -94, could you also speculate what could be the results of a vaccination campaign, like using live attenuated Qdenga (Takeda's Vaccine) on the incidence?
4) Line 283 It was not only the lack of demand (based on horrendous outcomes) but also the decision of the company to not produce anymore a dangerous vaccine. Please reformulate considering this.
Corrections:
A) Since the format is “Review”, you do not need to include the section Methods, indeed, it is irrelevant here, please delete the whole section.
B) Correct in line 275 “live vaccines” to “live-attenuated vaccines”.
C) In lines 275-276 you speculate that: “it’s reasonable to consider CYD-TDV in the recommended age group in endemic areas in those who have a laboratory evidence of prior dengue infection”. Do you know that this vaccine is not recommended at all, due to Vaccine-Associated Antibody-Dependent Enhancement (ADE)? This phenomenon in Dengvaxia (CYD-TDV, Sanofi-Pasteur) caused hospitalizations and deaths of children. Please reformulate considering this.
D) What you wrote in lines 279 -281 is not true, especially if you compare it with Dengvaxia. Reformulate considering current Vaccination campaigns in affected areas.
E) Finally, please explain more clearly why you wrote something like : “Routine donor and recipient screening is not recommended.” Justify in term of cost-benefit and not only thinking in US cases, please.
If all the corrections are implemented, the review will have green light for publication.
My best regards,
The Reviewer.
Author Response
Suggestions:
0) Please shorten as much as you can the length, maintaining only the most important information, if possible.
Response: The manuscript has been shortened.
1) In table 1 it will be interesting to incorporate information on which arbovirus cross-react with which (by ELISA, PCR, etc)? It will be interesting to add to this Table 1 in order to detect the infection in the patient and guess which virus is present based on geographical area or recent travels.
Response: We thought about it too but it might be cumbersome to add this information in the table, mainly because some of these viruses have not associated with infection in SOT and there is a lack of data on cross reactivity for these viruses. As a general rule, positive IGM is non specific and should be confirmed by neutralizing antibody assays ( if possible) but even neutralizing antibodies can cross react with various arboviruses. PCR and metagenomics are more specific but sensitivity varies widely. This is explained in the text but adding all this information in the table is cumbersome.
2) In line 83 please generalize to Aedes mosquitoes in endemic areas, instead of citing species.
Response: Corrected to Aedes species mosquitoes.
3) In lines 93 -94, could you also speculate what could be the results of a vaccination campaign, like using live attenuated Qdenga (Takeda's Vaccine) on the incidence?
Response: The following sentence is added:
“Although currently available dengue vaccination is not recommended in SOT, its implementation in routine immunization in high transmission settings is predicted to lower the incidence of dengue infection (especially DENV1 and DENV2 serotypes) in the general population and perhaps in SOT as well”. A new reference ( ref 29) is added to support this statement).
4) Line 283 It was not only the lack of demand (based on horrendous outcomes) but also the decision of the company to not produce anymore a dangerous vaccine. Please reformulate considering this.
Response: CYD-TDV is a dangerous vaccine only if its used in seronegative individuals. Hence it was recommended that it be used in individuals who had a serological evidence of prior infection. If used in individuals who never had dengue before and if they get natural infection later, it can cause severe dengue. This made it difficult in implementing it in routine immunization programs in endemic areas (since it required screening for prior infection). This is probably the reason why it was discontinued. The other dengue vaccine (TAK-003) does not require prescreening for prior infection and is the only WHO recommended vaccine at this time. Nonetheless we have rephrased this section to highlight this, especially since CYD-TDV is no longer manufactured.
Corrections:
A) Since the format is “Review”, you do not need to include the section Methods, indeed, it is irrelevant here, please delete the whole section.
Response: The other reviewer recommended keeping this section. We also think it will be helpful to the readers regarding the search process. Hence we decided to move this section to supplementary material.
B) Correct in line 275 “live vaccines” to “live-attenuated vaccines”.
Response: Corrected
C) In lines 275-276 you speculate that: “it’s reasonable to consider CYD-TDV in the recommended age group in endemic areas in those who have a laboratory evidence of prior dengue infection”. Do you know that this vaccine is not recommended at all, due to Vaccine-Associated Antibody-Dependent Enhancement (ADE)? This phenomenon in Dengvaxia (CYD-TDV, Sanofi-Pasteur) caused hospitalizations and deaths of children. Please reformulate considering this.
Response: We don’t believe the reviewer’s statement is entirely correct. CYD-TDV was recommended for certain age groups in endemic areas only if they had serologic evidence of prior dengue infection. It was NOT recommended in dengue naïve patients since natural infection following vaccination led to severe dengue infection in this group. The adverse effects mentioned by the reviewer were in seronegative individuals who received this vaccine. Anyway this is a moot point now as this vaccine is no longer manufactured. We have revised this section and removed the statement regarding CYD-TDV vaccination in pretransplant settings.
D) What you wrote in lines 279 -281 is not true, especially if you compare it with Dengvaxia. Reformulate considering current Vaccination campaigns in affected areas.
Response: We don’t believe the reviewer’s statement is correct. TAK-003 is effective against dengue serotypes 1 and 2 but its efficacy against serotypes 3 and 4 (especially in seronegative individuals) is not clear. Although screening for prior dengue exposure is not required, there is a concern that seronegative individuals who receive TAK-003 vaccination might develop severe dengue if exposed to natural DENV3 or DENV4. We have referenced the WHO position paper on this ( ref 29). For more information pls see the link below ( ref 29)
(https://iris.who.int/bitstream/handle/10665/376641/WER9918-eng-fre.pdf?sequence=1).
E) Finally, please explain more clearly why you wrote something like : “Routine donor and recipient screening is not recommended.” Justify in term of cost-benefit and not only thinking in US cases, please.
Response: This is now added. Routine screening is not done for most arboviruses due to low incidence in SOT (even in endemic areas) as well as the the possibility of having false positive screening tests which can lead to cancellation of life saving transplantation.
Reviewer 2 Report
Comments and Suggestions for Authors
This is a very interesting review providing a lot of information on a topic not well studied, although of increasing importance in public health. The outcomes of the review described in the method section must be available, even as a supplementary material to better understand the background information for the results/information presented in chapter 3.
A few specific and little edits will follow through the Editors.
Author Response
This is a very interesting review providing a lot of information on a topic not well studied, although of increasing importance in public health. The outcomes of the review described in the method section must be available, even as a supplementary material to better understand the background information for the results/information presented in chapter 3.
A few specific and little edits will follow through the Editors.
Response: The method section is moved to the supplementary section. The following information is added in this section: Only articles that reported arbovirus infection in solid organ transplant (SOT) were selected for the purpose of the review. We excluded articles that reported infection in cornea transplant only. The following arboviruses were found to be associated with SOT and are reviewed in the paper: dengue, Japanese encephalitis, chikungunya, yellow fever, zika, powassan virus, tick borne encephalitis virus, West Nile virus, eastern equine encephalitis virus, St Louis encephalitis, Jamestown canyon, cache valley virus, Crimean-Congo hemorrhagic virus, heartland virus, rift valley fever and usutu virus.
Other major changes:
The following information is added under “other arboviruses”
Although no infections in transplant recipients have been reported to date, the recent explosion of oropouche virus in regions of South America, particularly Brazil, raise concern going forward. Ref 167 added.
Round 2
Reviewer 2 Report
Comments and Suggestions for Authors
The revised manuscript addresses most of the comments, although a few minor clarifications and edits are still required.

Author Response
See tracked changes in the manuscript
Commented [R1]: Since this review does not provide criteria of inclusion/exclusion of articles, it is better to clarify that is a narrative review.
Title modified per reviwer’s suggestion
Commented [R2]: It would be good to provide a list of abbreviations since there are many in the document and it is not always straitforward to understand the meaning of the abbreviations. This list would facilitate the reading.
This section is added.
Commnet R3: What are the Orthomyxoviridae transmitted by arthropods ? I could not found any in Table 1 and no report of diseases caused by Orthomyxoviruses transmitted by arthropods in a quick literature search too.
Response: Table updated. Orthomyxoviridae family has Thogotovirus genus including thogoto, bourbon and dhori viruses . These have been reported to cause human infections. eg DOI: 10.3201/eid2105.150150
Commented [R4]: The paragraph on the methods has been moved to supplementary files, but still miss some information regarding the selected articles such as how many publications were retrieved and how many were used, for each combination of SOT and disease. It would be helpful to understand how large is the available information.
Response: One of the reviewers suggested removing this section entirely since this a narrative review. But agree with keeping this section. However, since this is not a systemic review with stringent inclusive and exclusive criteria , we did not keep track of the articles that we searched. Moreover, just mentioning how many articles were found and not going into details of how many were duplicate and how many were excluded, is not going to be meangiful. All relevant articless have been referenced. We think the current format should be Ok. We also added a few viruses in the search criteria ( to include orthomyxoviridiae )
Commented [R5]: The 2 main vectors of arboviruses are well Aedes aegypti and Aedes albopictus, the names of thoses species must remain in the text, because Aedes genus include hundred of species, not vectors.
Response: Corrected
Commented [R6]: The use of wolbachia infected mosquitoes is not yet recommended by WHO and prevention against mosquito bites as well as elimination of mosquitoes breeding sites are the more common vector control practices.
Response: Although not recommeded by the WHO, it has been used with success. So we think its best to keep this section so that the readers are aware of its potential use. However we modified the sentence to highlight that its not yet recommended by the WHO.
Commented [R7]: The main vectors of JEV are Culex mosquitoes with very different behavior than the dengue vectors. Mosquito control against the Culex include elimination of breeding-sites, use insecticides and impregnated bednets because Culex are biting at night.
Resposne: Not sure that there are drastic differences in mosquito control startegies between Aedes and culex. (except using mosquito nets during daytime vs night time) Anyway added a few sentences for clarity
Commented [R8]: Although most of the arthropod-borne flaviviruses are asymptomatic, Yellow Fever is mostly symptomatic, even if the fever is of short duration.
Response: We removed this sentence
Commented [R9]: The reference(s) is (are) missing for the duration of 3 and 2 months respectively.
Response: Ref added
Commented [R10]: What is written here is the contrary with an incidence of 1.22 in less than 10 years and an incidence of 0.22 for above 70 years. Please check and correct.
Resposne: Its now corrected
Commented [R11]: What is wbc ?
Resposne: wbc is “white blood cell” and now explained
Commented [R12]: Checking for mosquito bites is unrealistic and cannot be done, worldwide.
Resposne Agree. The phrase “ or recent mosquito bite” is removed.
Commented [R13]: What is NAT ? Not spelled above.
Response: Its nucelic acid amplification test and is now mentioned in section 2.8.3
Commented [R14]: Since IgM are proof of a recent infection, it is very surprising that organ donation can be considered from a donor who had recent WNV infection ???
Response: This is the recommendation directly from the United States Organ procurement and Transplantation Network (OPTN) ad hoc Disease Transmission Advisory Committee ( ref 150). Ig M does not always imply active infection since this test can linger on positive for months. So during donor sreening, blood WNV PCR is checked first and if positive donation defered and 28 days later blood PCR and IgM chekced. If blood PCR neg after 28 days, its unlikely to transmit infetcion. The presence of IgM with negative PCR confirms a recent but not active infetcion ( recovery from recent infection). If both IGM and PCR neg at this time, it implies that the initial PCR was false positive. This has already been explained in the text.
In the conclusions ectuon the reviewer suggested adding the following pharse after poor specificity. “ between the different families. Nevertheless positive IgM are proof of a recent infection with one virus of the family and thus can help to eliminate potential donors.”
Response: Pos IGM is tricky. its not only non-specific between diffent arboviruses, it can be poistive even in the absence of an arobiral infetcion. PRNT is more specific in this regard. Also Ig M can remain positive for months and does not imply a recent infetcion all the time.Hence its best not to use thsi pharse